# Influence of Liquid-to-Solid and Alkaline Activator (Sodium Silicate to Sodium Hydroxide) Ratios on Fresh and Hardened Properties of Alkali-Activated Palm Oil Fuel Ash Geopolymer

**DOI:** 10.3390/ma14154253

**Published:** 2021-07-30

**Authors:** Shi Ying Kwek, Hanizam Awang, Chee Ban Cheah

**Affiliations:** School of Housing, Building and Planning, Universiti Sains Malaysia, Gelugor 11800, Penang, Malaysia; sheily1999@hotmail.com (S.Y.K.); cheahcheeban@usm.my (C.B.C.)

**Keywords:** geopolymer, industrial waste, POFA, alkaline activator, sustainability

## Abstract

Malaysia is one of the largest palm oil producers in the world and its palm oil industry is predicted to generate a large amount of waste, which increases the need to modify it for sustainable reuse. The green geopolymers produced from industrial waste can be a potential substitute for cementitious binders. This type of polymer helps reduce dependency on cement, a material that causes environmental problems due to its high carbon emissions. Palm oil fuel ash (POFA) geopolymer has been widely investigated for its use as a sustainable construction material. However, there is still uncertainty regarding the total replacement of cement with POFA geopolymer as a binder. In this study, we examined the effects of different material design parameters on the performance of a POFA-based geopolymer as a building material product through iterations of mixture optimisation. The material assessed was a single raw precursor material (POFA) activated by an alkaline activator (a combination of sodium hydroxide and sodium silicate with constant concentration) and homogenised. We conducted a physical property test, compressive strength test, and chemical composition and microstructural analyses to evaluate the performance of the alkali-activated POFA geopolymer at 7 and 28 days. According to the results, the optimum parameters for the production of alkali-activated POFA paste binder are 0.6 liquid-to-solid ratio and 2.5 alkaline activator ratio. Our results show that the use of alkali-activated POFA geopolymer is technically feasible, offering a sustainable and environmentally friendly alternative for POFA disposal.

## 1. Introduction

Geopolymers have become a green technology that is continuously developed by researchers as a cement replacement due to their similar binder characteristics and mechanical properties [1,2]. Using geopolymers instead of cement has benefits such as environmental friendliness (reduced use of cement), reduced CO_2_ emissions_,_ and reduced energy consumption [3,4,5]. Given the high demand for industrial construction materials, finding ways to prevent the scarcity of natural resources, aside from considering the usage of waste material in the construction industry, has become a main concern for the industry. The use of geopolymers is one way to reduce waste, especially industrial waste, and to explore innovative methods for waste management [6]. A mind-set of ‘turn waste into wealth’ is causing the current shift towards environmentally friendly development [7].

Geopolymerisation is a reaction (inorganic polycondensation) that involves a three-dimensional zeolitic framework to form geopolymer cement [8]. The mechanism of geopolymerisation hardening occurs when aluminium and silicate oxides existing in any raw material react with alkali polysilicates. The type of material, its fineness, size, and the activator used are the main influencing factors of the geopolymerization process [9]. Alkaline-activated geopolymers generated from industrial byproducts using an aluminosilicate reaction, such as fly ash, bottom ash, slag, and others, have been studied previously. In general, an aluminate silicate reaction, calcium silicate hydrate (C-S-H) reaction, and the subsequent formation of geopolymeric gel are the geopolymer hardening processes that bridge the gaps within the matrix [10]. The activation of the alkaline activator with kaolin involves three stages: destruction, polymerisation, and stabilisation (Figure 1). The rate of reaction to improve hardening can be increased under an optimum temperature or curing process. The geopolymerisation of alkaline-activated POFA shows an almost similar mechanism (Figure 1), in which the polycondensation of both hydrolysed aluminate and silicate is the main reason for the formation of C-S-H gel and the coexistence of geopolymeric gel during hardening of the geopolymer binder. All the elements contribute to the creation of connecting bridges within matrix gaps.

Sodium hydroxide (NaOH) and sodium silicate (Na_2_SiO_3_, water glass) are used as alkaline activators to activate geopolymers. NaOH dissolves aluminosilicate, while Na_2_SiO_3_ supports NaOH as a binder, plasticiser, or dispersant [12]. Based on preliminary laboratory work, it was shown that NaOH alone cannot boost the compressive strength of geopolymers [13]. Studies have shown that the performance of geopolymer binder depends on several parameters, including the concentration of activator, activator ratio, and liquid to solid ratio, among others. Most geopolymer mixes show good compressive strength at an alkaline activator ratio of 2.5 [12,14,15]. Kastiukas et al. [16] found a smaller effect of NaOH in the alkaline activator composition compared to the content of Na_2_SiO_3_ on mining waste_._ Varying the Na_2_SiO_3_ content affects the workability, setting time, and compressive strength of the activated mixture. Each geopolymer made from different raw materials will be sustained from the suitable activator ratio. The properties of the materials used affect the demand for activator. According to a study by Xu and Van Deventer [17], a fly ash to liquid ratio of three is needed to promote geopolymerisation. Wan et al. [18] reported that the strength of geopolymers is affected by the NaOH concentration, with 12 M the optimal level. Among cementitious properties, high strength can be produced by alkali activation, which depends on the SiO_2_ and Al_2_O_3_ content [19].

Palm oil fuel ash (POFA) is the industrial waste generated from the palm oil industry. POFA has a pozzolanic reactive property and thus can be used as a partial replacement for cement [20,21] or as part of a binary mixture with other aluminosilicate materials (commonly fly ash, granulated blast furnace slag, and rice husk ash) to form geopolymers. In most previous studies, a low content of POFA was used as a supplementary material to produce geopolymer. Salih et al. [22] discovered that the incorporation of slag with alkali-activated POFA improved the final product’s strength, as the slag product was crystalline and POFA was predominantly amorphous. Hussin et al. [23] found that replacement with 30% ground POFA results in higher-strength concrete compared with ordinary Portland cement concrete at a later age. Hussin et al. [24] used ground POFA in aerated concrete to show that POFA has the potential to produce different types of concrete, and it also exhibited better performance against drying shrinkage. Pourakbar et al. [25] produced bricks with adequate compressive strength using POFA. The curing method used affects the performance of POFA geopolymers. Polymers can gain a strength of up to 10 MPa with oven curing at a 0.55 ratio, 10.8 MPa when cured at 60 °C, and 13 MPa at a 0.5 ratio and 30 °C [26,27,28]. POFA geopolymers under an ambient curing regime were found to achieve a compressive strength of 31 MPa [9]. Karim et al. [29] claimed that POFA containing a high amount of silicon dioxide in amorphous form can react with calcium hydroxide generated during the hydration process to produce more C-S-H gel compound, as shown by the following formula [30]:2S + 3CH = C_3_S_2_H(1)

The final products in a pozzolanic reaction cannot be distinguished from the primary cement hydration. As a result, the products contribute to the strength and other properties of hardened cement paste and concrete [11]. Given the rapid development of the construction sector, concerns regarding issues including the sources of building materials used, the effect of the production of construction goods, and sustainability, have caught the attention of major industry players. The use of green construction materials for construction work is essential. Thus, alternative solutions are being investigated to promote environmentally friendly building materials as a replacement for cement, sand, or aggregate. Given the ongoing depletion of natural resources and huge amounts of unwanted waste being produced by different industries, possible solutions should be developed to produce new materials that benefit the construction industry. Creating alternative products is essential to solve the current issues facing the construction industry.

In this paper, we focus on the performance of alkali-activated POFA geopolymer as a binder or cementitious construction application material. We investigated the influence of two parameters, namely, the liquid-to-solid ratio (L/S) and alkaline activator ratio, on the physical and mechanical strength of alkali-activated POFA geopolymers at room temperature. In this case, the liquid is the activator solution and the solid is the geopolymer solid (POFA) in the liquid/solid (L/S) ratio. For the alkaline activator ratio, we consider the composition of Na_2_SiO_3_ to NaOH. In general, geopolymers can be cured in a temperature range from 60 to 80 °C, depending on the nature of the materials used. This study was conducted to achieve a practical and preferable in situ cast application without high-temperature curing. Moreover, with growing interest in the production of geopolymer artificial lightweight aggregates, the POFA-based geopolymer’s behaviour as a substitute for production needs to be tested. Our findings may promote further investigations of the application of sustainable geopolymers in concrete and as a raw element for commercial use and construction work.

## 2. Experimental Method

### 2.1. Materials

POFA is the residue generated in the form of ash via palm oil husk combustion at high temperatures. The raw material was dried at 100 ± 5 °C for 24 h and then sieved through a 300 µm mesh and ground to obtain a fine POFA powder with a particle size of less than 45 µm. Figure 2 shows that the POFA contained 90% (d_90_) fine particles measuring 35 µm. The grinding process can change the structural properties of particles from an irregular to a spherical shape in order to form a homogenous powder [30]. Raw POFA cannot be used without sieving and grinding, given its relation to the porous structure and high demands for alkaline activator that attains a low compressive strength [31]. Figure 3 shows the different shapes of POFA obtained from the grinding process. Unground POFA showed an irregular and porous cellular shape (Figure 3a), but after grinding it had rough surfaces with spherical particles (Figure 3b). Salih et al. [11] reported that POFA materials exhibit a fine particle size in a homogenous powder with an increased rate of alkaline activation in reaction after grinding.

X-ray fluorescence (XRF) analysis was performed to determine the oxide composition of POFA [32]. The principal oxides included for POFA were silica (SiO_2_), calcium oxide (CaO), and potassium oxide (K_2_O), accounting for 54.98%, 10.77% and 9.50% of the total composition, respectively. Given its high silica content, POFA has been promoted as a binder [33]. Compared with the oxide compounds that exist in cement, CaO represents the largest portion of POFA, followed by SiO_2._ We focused on determining the possibility of POFA reacting with an activator additive to achieve cementitious properties. Our findings can also indicate whether or not POFA should be categorised as a toxic waste material [34]. X-ray diffraction revealed that the matrix of POFA contained predominantly quartz (SiO_2_), kyanite (Al_2_O(SiO_4_)), and gehlenite (Ca_2_Al(AlSiO_2_)) as the primary crystalline phases of the material (Figure 4). It is important to define the amorphous properties of POFA. As for the geopolymer, the amorphous alkaline aluminosilicate gel is the main product formed at the end of the process [35].

The alkaline activator for the activated POFA geopolymer was a combination of 12 M NaOH and Na_2_SiO_3_. NaOH was in the solute form of formosoda-P NaOH (reagent grade with a 99% purity), whereas Na_2_SiO_3_ was in liquid form. Na_2_SiO_3_ consists of SiO_2_ (30.1%), Na_2_O (9.4%), and water (60.5%), and has a specify gravity of 1.4. NaOH is essential for geopolymerisation because the presence of hydroxide ions activates the dissolving Si and Al ions in the gel phase to form an Si-O-Al monomer [36]. Both alkalines were of industrial grade and exhibited a stable performance. Although the concentration of alkaline used affects the strength of geopolymer, compared with other parameters its effect is less significant. The NaOH molarity we used was based on the recommendations of prior related studies [37,38]. The concentration of alkaline was directly proportional to the compressive strength present. However, this parameter decreases the strength of the material, as the molarity exceeded the limit of Na ions to react with elements within the raw material used. A 12 M concentration NaOH solution was prepared by dissolving a total of 480 g of NaOH solute per litre of water. An alkaline solution was prepared 24 h before it was used ensure a full exothermic reaction before being used in mixing to prevent the flash set of paste samples due to the heat of hydration [39].

### 2.2. Mix Design and Procedure for Alkali-Activated POFA Paste

Two parameters were determined for the study: L/S ratio was determined first (Table 1) followed by the ratio of the alkaline activator (Table 2). The mixture proportion of alkali-activated POFA geopolymer paste was finalised based on the preliminary experiments. The unground POFA had a lower compressive strength than the ground POFA due to its porous microstructure. The mixture was prepared by pouring weighed POFA and mixing it simultaneously with the alkaline activator solution based on the mix proportion specified by the L/S ratio. Mixing lasted several minutes, with a pause to scrub the mixture stick on the mixer, followed by another 5 min of mixing. A flow test was carried out to determine the flowability and workability of the mixture. The flow table apparatus used for the test complied with specifications in ASTM C230 [40] and the standard testing methods stated in ASTM C 109 [10]. The mixture was cast in a 50 mm cube under vibration as per ASTM C109 [10] to remove the air inside the paste, and then this was left for a day. The alkali-activated POFA geopolymer was demoulded and plasticcured at room temperature (25 ± 5 °C). The same procedure was repeated for the alkaline activator ratio parameter with a constant optimisation L/S ratio.

### 2.3. Testing Methods

Changes in the properties of the alkali-activated POFA geopolymer were evaluated to ensure the mixture’s uniformity during the mixing process. Physical tests included density and flow tests (ASTM C 1473) [41], and a mechanical compressive test was also used (ASTM C 109) [10]. Three samples were tested for each mixture at 7 and 28 days after curing. 

The initial and final setting times were based on ASTM C191 [42]. Standard consistency was not measured due to the flowability requirement and the L/S parameter. The optimum ratio of the liquid was determined based on the flowability and strength. Fourier transform infrared (FTIR) analysis was carried out on three groups (control, minimum strength, and maximum strength) to investigate the composite compounds and the functional group of the mixtures. Each sample was ground into powder and compressed to a disc form for testing. The sample was scanned through IRPrestige-21 at wavelengths ranging from 400 to 4000 cm^−1^ by IRsolution. However, XRD analysis revealed less apparent formation of amorphous materials from 37 to 70 °C, as only the intensity change of XRD spectra was visible for advancement of the geopolymerisation reaction [43]. Thus, confirmation of amorphous gel formation can be determined from SEM characterization [44]. Scanning electron microscopy (SEM) and energy dispersive X-ray (EDX) analysis were also conducted to differentiate the alkaline element’s effect on the different compositions of the alkali-activated POFA geopolymer. A small cube formed from paste was used for testing. The small dry cube surface was polished and coated with gold using a Quorum Q150TS metallic coating machine (Quantum Design GmbH, Darmstadt, Germany). The coating process was performed using a constant sputter current of 120 mA, sputter time of 60 s, and a tooling factor of 2.28. For microstructural assessment, a constant acceleration voltage of 15 kV was used as the electron beam intensity in a Quanta FEG 650 machine. A magnification factor of 5000× was applied to define the morphological parameters. These tests were performed after 28 days of curing. 

## 3. Results and Discussion

### 3.1. Effect of L/S Ratio

The flow was determined to identify the workability of the paste mix. As the L/S ratio increased from 0.50 to 0.85 by increments of 0.05, the flow increased (Figure 5). This finding can be explained by the high liquid content at a high L/S ratio, which resulted in a high flow [45]. At an L/S ratio below 0.50, the sample could not be mixed and vibrated to form a homogenous paste due to the limit of the solvent. The dissolution of POFA was possibly inhibited by the activator, given the incomplete geopolymerisation process. With a totally alkaline activator used for the mixture, the required amount of liquid corresponded to the flowability of the alkali-activated POFA geopolymer to achieve the desirable lubricating effect. The irregular and porous nature of POFA increased the demand for the alkaline activator. However, at ratios above 0.85, the amount of alkaline activator was excessive and caused high flowability. This condition slowed down the setting time and caused an imbalance in the elements required to undergo the chemical reaction within the mixture [46].

The alkali-activated POFA geopolymer density decreased as the L/S ratio increased (Figure 5). This may be due to the condensation process during the polymerisation reaction. With low alkaline activator content, the reaction of POFA geopolymer retarded which limits the complete hydration reaction (Sinduja et al. [47]). The alkali-activated POFA geopolymer paste had a density of up to 1700 kg/m^3^, which was slightly higher than that of fly ash geopolymer [47], but lower than that of Portland cement paste [48]. The change in density at different L/S ratios was influenced by the geopolymerisation and curing conditions, given the involvement of the dehydroxylation and crystallisation phases as well as the curing period. The optimum liquid ratio promoted the reaction between the POFA and activator.

As the curing age of the sample increased, its density decreased due to reduced water content from the hardened geopolymer. This was due to the complete reaction and evaporation of the sample’s water content. This finding indicates that the initial stage of reaction was faster than the latter stage. Therefore, a slight reduction in density occurred during the latter stage.

According to Awaluddin et al. [13], compressive strength increases with NaOH concentration and curing temperature but decreases with the L/S ratio. The sample with highest workability had the lowest strength (Figure 6). This finding can be explained by the findings of a study by Salih et al. [7], in which two values were used to investigate the effect of L/S on the performance of an alkali-activated POFA geopolymer. The results showed that a high liquid content could be attributed to low strength due to the void volume within samples. At L/S ratios of 0.76 and 1.0, strengths of approximately 32 and 24 MPa were observed after 28 days of curing, respectively. Thus, the curing conditions caused differences in the results. In the present study, higher strength was achieved at lower L/S ratios.

Based on the work of Hairi et al. [14], despite the decreased L/S ratio, the use of pastes with a L/S ratio below 0.55 was not feasible because the paste rendered the moulding procedure difficult and inefficient. The optimum range of L/S ratios had to be defined to achieve sufficient wetting of solid grains without excessive water. The lowest compressive strength was obtained at an L/S ratio of 0.85. The low compressive strength was due to the high content of the liquid portion of the solute. Moreover, the high proportion of alkaline activators resulted in high levels of OH ions in the mixture, which indirectly weakened the bonding structure during the geopolymerisation reaction [49]. Therefore, an L/S ratio of 0.6 was selected to achieve the desired flow and compressive strength of the alkali-activated POFA geopolymer.

### 3.2. Effect of Alkaline Activator Ratio

A decreased flow trend was observed as the alkaline activator ratio increased (Figure 7). Alkaline activator ratios of 0.5 and 3.5 were not used due to the unworkability of samples. At an alkaline activator ratio of 0.5, the POFA paste set before pouring and vibrated inside the mould, preventing it from being moulded. This was due to the excess NaOH content, which caused a rapid polymerisation reaction. At an alkaline activator ratio of 3.5, the POFA paste was too stiff to mould. Flowability depends on the alkaline activator ratio, whereby Na_2_SiO_3_ acts a as suspension liquid for geopolymers [50]. The sample with NaOH only showed excellent flowability.

Figure 6 shows the density of the alkali-activated POFA geopolymer. The sample densities ranged from 1658 to 1741 kg/m^3^ for the binary alkaline activator. The sample with NaOH only had a lower density than the sample with Na_2_SiO_3_ and NaOH. Several factors affected the density of the samples including the dehydroxylation and crystallisation phases that form during the geopolymerisation process, bonding, and the presence of a hydroxylase group (–OH) [50]. Our results show that the alkaline activator ratio of 1.5 had least influence on density. This might be due to an incomplete reaction to form the polymer bond within the mixture. Although satisfactory compressive strength was observed at ratios of 1.5 and 2.5 in several studies [51], we found that the 2.5 ratio provided better strength. This might be due to the desirable content of Na_2_SiO_3_ that allowed the reaction to proceed. The geopolymer had a high density because of the high Na content in the alkaline activators, which contributed to the high reactivity, leading to denser samples. Therefore, the mixture of alkaline in this proportion was suitable to react with POFA and undergo effective geopolymerisation. In addition, the density slightly decreased for the majority of samples. This was attributed to the reduced geopolymerisation process in the latter stages and water loss due to evaporation during the reaction.

The initial and final setting times were indicated for the mixture for the single and binary alkaline activators. Based on a study by Salih et al. [11], the initial and final setting times for Portland cement were 50 and 195 min, respectively. Figure 8 shows the setting times for the alkali-activated POFA geopolymer at different alkaline activator ratios. It also shows that the mixtures required considerable time to set. The use of POFA as the main material prolonged the setting reaction. Our results show a decline in the setting time as the Na_2_SiO_3_ content increased. The mixture with a 3.0 ratio of Na_2_SiO_3_/NaOH required the shortest setting time (Figure 8). According to ASTM C150 [52], the initial minimum setting time was 45 min, and the maximum setting time was 365 min. For all mixtures, the initial setting time for the majority of samples was not in the recommended range, but five out of six mixtures satisfied the maximum range value requirement. The mixture with NaOH only presented a prolonged setting time. As the Na_2_SiO_3_ increased, the silica content also increased, which accelerated the reaction. An extremely fast setting time affects the mixture’s properties because immediate reactions are retarded as the hydration process further progresses. The high NaOH content delayed setting. This was probably due to the limited capability of single binary alkaline activators during geopolymerisation. According to Tchakoute et al. [53], a metakaolin geopolymer had higher setting times as the NaOH content increased.

Figure 9 shows that the compressive strength increased as the alkaline activator ratio increased due to the content of Na_2_SiO_3_, which acted as the coagulate to accelerate dissolution during polymerisation [25]. Moreover, Na_2_SiO_3_ can be the intermediate substance or plasticiser for the polymerisation reaction. The ideal alkaline ratio for the majority of geopolymers is 2.5. In this study, the alkali-activate POFA geopolymer was formed at the alkaline activator ratio of 2.5. The sample with an alkaline activator ratio of 2.5 produced most geopolymeric gel with the intervening material. The compressive strength marginally declined when the ratio reached 3.0. This result was similar to a previous study, in which the compressive strength also decreased beyond a ratio of 3.0 [48,54]. Excessive Na_2_SiO_3_ content inhibits the geopolymerisation reaction because Na ions might prevent some of the precipitation phase from contacting the POFA and alkaline activator [55]. The alkaline activator NaOH in the POFA geopolymer resulted in the lowest strength in this study. The high concentration of OH ions caused early hydration gel precipitation, thus hindering geopolymerisation and leading to the low strength of the geopolymers. This result was the same as that found by Hwang and Huynh [56].

Curing was performed at ambient temperature in this study, which consumed lower amounts of energy. This was aimed at improving sustainable practices, since the performance was still within the acceptable range for construction use. In the work of Shubbar et al. [57], the use of a complete POFA geopolymer achieved a strength of 18 MPa after 28 days of curing, which was performed at 65 °C in an oven for 24 h. Our results were nearly the same as those of the previous research [57]. The oven curing period also affected the performance of the POFA geopolymer. Prolonged curing resulted in a cracked appearance, which reduced the compressive strength [58].

Efflorescence occurred on the surface of the geopolymer cube after 28 days of curing at an alkaline activator ratio of 1.0 (Figure 10). However, at different alkaline ratios, various proportions of efflorescence were recorded with different NaOH and Na_2_SiO_3_ contents. The results illustrate that additional NaOH content increases the potential for the occurrence of efflorescence. This condition results from excessive sodium ions (an alkali element) in the mixture. The sodium ions react with atmospheric carbon dioxide and cause white spots to develop. However, the presence of efflorescence did not directly affect the strength of the geopolymer. Excessive sodium ions were bound inadequately in the geopolymer gel’s nanostructure, reducing the mobility of alkalis, which were almost entirely leached without reducing compressive strength [46]. As a result, the lowest ratio of alkaline activator exhibited the lowest strength (Figure 9).

### 3.3. FTIR Analysis

The chemical bond and mechanism of bonding of geopolymers was investigated using FTIR spectrum results. Wen et al. [59] showed that FTIR analyses not only provide valuable information related to the nature of bonding for each of the test samples, but also offer a simple technique to gather test data that substantiate the mechanism driving geopolymerisation. The FTIR spectrum shown in Figure 11 shows similar spectrum profiles for the three mixtures. The reaction occurred and the peak intensity varied based on the effect of different alkaline activator ratios, that is, the control mix (NaOH), optimal mix (Na_2_SiO_3_/NaOH = 2.5), and weakest mix (Na_2_SiO_3_/NaOH = 1.0). The presence of Na_2_SiO_3_ affected the hydroxyl and carbonyl groups in the mixture [46]. The Si-O-T peak was used to define the geopolymer’s presence, where T represents tetrahedrally bonded Si or Al at wavelengths in the range 800–1300 cm^−1^.

Figure 11 shows the main peak within the range of wavelength, showing evidence of geopolymerisation reaction products. The main bands associated with POFA and Na_2_SiO_3_/NaOH = 1.0 were 465, 671, 779, 1024, 1429, 1655, and 3248 cm^−1^. For the mixture with Na_2_SiO_3_/NaOH = 2.5, bands were detected at 465, 673, 779, 1020, 1431, 1655, and 3426 cm^−1^. The peaks for the mixture with NaOH only were located at 457, 679, 777, 1015, 1456, 1653, and 3246 cm^−1^. The peaks at wavelengths in the range 400–500 cm^−1^ were attributed to the symmetric stretching vibration of Si-O-Si, which represents quartz [60]. A Si-O-Si bending vibration was observed at wavelengths in the range 680–800 cm^−1^. At wavelengths of 1050–1095 cm^−1^, the content of kyanite was consistent with that present in POFA (Figure 2). The intensities of bands were assigned to the stretching of –OH and bending of H-OH at wavelength ranges 1620–1640 and 3000–3700 cm^−1^.

For the alkali-activated POFA geopolymer with an activator ratio 1.0 (Figure 11a), the peak band trend showed no substantial difference from that of the mixture with an activator ratio of 2.5, but presented a slight peak difference compared with the mixture with a single activator. The addition of Na_2_SiO_3_ enhanced the shift towards intermediate wavelengths, and the visible bands were within the range of the representative peak. The intensity of the carbonate peak centred around 1429 cm^−1^ was reduced when Na_2_SiO_3_ content increased.

Figure 11b shows that the wavenumber located at 1020 cm^−1^ indicated a progressive reaction given the rich silica gel. As mentioned by Nadzini et al. [46], this result was consistent with the formation of sodium aluminosilicate gel. This visible band indicated the presence of the geopolymeric gel, leading to a high compressive strength due to the high concentration of Si-O-Al bonds. The geopolymeric gel had a low content of carbonate compared with the mixture with a single alkaline activator (high ratio of NaOH). Moreover, a 1655 cm^−1^ band was located slightly beyond that of the control mix (mixture with NaOH) because of the reduced intensity of the band assigned to the stretching of –OH due to the reduction of the NaOH ratio in the mixture. The visible sharp band at 3426 cm^−1^ is indicative of H-O-H bending. This peak appeared due to the binder hydration process and the presence of hydration products.

For the mixture shown in Figure 11c, the bands at 457 and 777 cm^−1^ fell within the bending vibration Si-O-Si band range. The asymmetric stretching vibration of the Si-O-T band shifted to a higher frequency wavelength, indicating a reduced polymerisation framework formation. The peak band at 1456 cm^−1^ represents the asymmetric carbonate strength. Usually, this finding is observed for NaOH-rich geopolymers [61] due to sodium carbonate formation. The distinct bands at 1653 and 3246 cm^−1^ are associated with a H-O-H bending vibration and a stretching vibration of O-H, respectively. The intensity of the stretching vibration of O-H decreased compared with the mixture with a combination of binary activators. The reduction in the amount of weakly bonded molecules is indicative of low reactivity and formation of geopolymeric products. This finding was due to the weakly bound water molecules, which were trapped in large gaps between the geopolymerisation products [62]. Table 3 summarises the characteristic absorption peaks of the FTIR spectrum.

### 3.4. Morphological Analysis by SEM and EDX

SEM/EDX analyses were performed to examine the phase characteristics and microstructural development of the geopolymer pastes. The SEM image of the POFA shows its irregular shape with round edges [46]. POFA also consists of porous cellular surface particles. The grinding process effectively reduced the particle size of POFA and achieved a homogeneous powder with a lower variation of particle sizes. Moreover, based on a study by Jamo et al. [63], the ground POFA was expected to exhibit higher reactivity and strength development due to its higher specific surface than the untreated POFA. The treated POFA particles were in a crushed form and spherical with a rough surface. Preliminary work performed using ground and unground POFA provided evidence that the ground POFA geopolymer exhibited higher compressive strength. The SEM image for raw POFA (Figure 3) indicated the reaction output for the activated POFA geopolymer. 

Figure 12 shows the microstructure of the alkali-activated POFA geopolymer paste that reacted with the alkaline activator at 28 days after curing. Figure 12a shows that the binder matrix consisted of fragmented particles, which indicates that the formation of aluminosilicate gel (main geopolymer product) was insufficient to form a fully interconnected microstructure framework. Furthermore, an incomplete reaction existed in the sample. This observation is in accordance with the mechanical behaviour of the paste samples. Figure 12b shows the rigorous formation of geopolymer matrix particles, which would probably fill the void among the final products. This explains why the mechanical test result indicated that the alkali-activated POFA geopolymer with an alkaline activator ratio of 2.5 achieved the highest strength at 28 days after curing. However, some microcracks were detected, which possibly occurred during the testing preparation. Figure 12c shows that the sample with NaOH only, the polymerisation reaction of which was incomplete, had a significant number of unreacted POFA particles. The slow solid geopolymer formation resulted in a low strength (5.54 MPa). The samples shown in Figure 12a,b formed a denser microstructure compared than those with NaOH only (Figure 12c), which was relatively more porous. A discernible formation of the crystalline compound within the binding gel was observed in the NaOH-activated binder system. The POFA activated by NaOH and Na_2_SiO_3_ showed a heterogeneous gel formation within a porous microstructure. The geopolymer pastes exhibited high porosity and permeability. 

EDX analysis was performed to identify the elements that constituted the reaction products observed in Figure 12 [53]. Microstructural investigations by EDX analyses showed that the glassy matrix consisted mainly of Na and Si, which are the essential constituents of geopolymer gels [64]. As identified by the EDX technique, these elements were compatible with the chemical compositions presented in the precursor material, POFA, which had a low content of Al_2_O_3_ and high content of SiO_2_. Na was detected in the binder phase of the alkali-activated composite; however, Na was not detected in POFA because it was a major element of the alkaline activator solutions (NaOH and Na_2_SiO_3_). 

A similar observation was reported by Salih et al. [11], in which the Na found in the binder phase was contributed by the alkali activator, which consisted of Na_2_SiO_3_ and NaOH. Morphologically, this phase was similar to that of the C-S-H phase, which is produced from Portland cement hydration. C-S-H gels exist with almost the same reaction from the hydration of Portland cement, but the difference lies in the lower Ca/Si atomic ratios (0.6–1.0) [46]. For this study, the Ca/Si ratio of C-S-H coatings for geopolymer paste with single or binary activators was analysed based on the EDX results. All the mixtures were categorised under medium geopolymerisation, because the POFA contained low calcium content. In theory, the strength of the alkali-activated binder corresponded to the Si/Al ratio (due to Si–O–Si bonds), where the strong Si–O–Al and Al–O–Al bonds would be amplified with the increase in Si/Al ratio [44]. The high SiO_2_/Al_2_O_3_ ratio was reported as a factor preventing the strength from improving in alkali-activated binder systems.

The EDX results shown in Figure 12a show a high Si/Al ratio of 15.24. The reaction between POFA aluminosilicate materials and alkaline activators (NaOH + Na_2_SiO_3_) continued, and excess Si/Al ratio was observed due to the depletion of Al content in the POFA material, which affected the mechanical performance of the alkali-activated POFA geopolymer. The complete reaction products formed in the geopolymer paste with an Na_2_SiO_3_/NaOH ratio of 2.5 (Figure 12b), as the raw POFA substances reacted with the alkaline activator and produced geopolymer products. An Si/Al ratio of 23.25 contributed significantly to the strength of the alkali-activated binder system. According to Qureshi and Ghosh [65], a relationship exists between the Si/Al ratio and compressive strength, as a high ratio of Si/Al leads to a highly crystalline matrix. In addition, the Si/Al ratio had initial and later effects on the microstructure of the alkali-activated binder. The hydration gel also helped it develop until it achieved the maximum strength for POFA. Furthermore, C-S-H formation may have partially contributed to the strength of the aluminosilicate phase or its development with a low Si/Al ratio. This explains why the sample with an alkaline activator ratio of 2.5 can achieve high strength, despite its Si/Al ratio not being the highest. The EDX analysis in Figure 12c showed a low Si/Al ratio of 12.82. The low ratio of Si/Al will result in loss of compressive strength. A low Si/Al ratio can result in low strength of the aluminosilicate compound, as the final crystalline phase is increased.

## 4. Conclusions

The nature of alkaline-activated POFA geopolymers depends on different parameters. The performance of the binder products formed can be determined from laboratory tests and by studying the microstructure and mineralogy of the samples. Our results were used to assess the alkali-activated POFA geopolymer application as a sustainable binder in artificial aggregate production.

Our findings follow:The L/S and alkaline activator ratios are the most important parameters that influence the physical and compressive performance of alkali-activated POFA geopolymer paste.At different L/S ratios, the alkali-activated POFA geopolymer’s flowability was inversely proportional to the compressive strength due to the excess liquid content, where it provides extra solution to react with the POFA for hydration.For the Na_2_SiO_3_/NaOH ratio, the alkali-activated POFA geopolymer’s flowability was directly proportional to compressive strength. A suitable alkaline activator ratio will provide an alkaline environment and inhibit polymerisation. However, when the ratio of alkaline activator mixture exceeds three, it retards the reaction. This is due to the excessive silicate ion content in the geopolymerisation reaction. Moreover, the formation of efflorescence obviously affects the performance of the alkali-activated POFA geopolymer, which occurs as result of a relatively high proportion of NaOH in the activator.In the early stages, the alkali-activated POFA geopolymer paste that was cured at room temperature had a lower strength than samples cured under general geopolymer curing conditions. However, this paste still reached a comparable strength later on in the experiment. This alkali-activated POFA geopolymer can attain binder characteristics for use as construction materials, such as cement replacement.C-S-H binding gel and sodium aluminosilicate hydrates are dominant in the POFA activated by alkaline.Through observation by SEM and EDX imagery, we found that the compressive strength determined the performance of the samples made by using different mixtures. There was a large difference between Si/Al ratios with difference activator ratios defined from EDX, which concluded that high Si/Al ratios lead to a highly crystalline matrix. However, FTIR spectra indicated the existence of a C-S-H formation.The activator POFA has properties that give it the potential to be used as a sustainable building material. Its cementitious characteristic as a binder would provide the ideal outcome for the construction industry.

## Figures and Tables

**Figure 1 materials-14-04253-f001:**
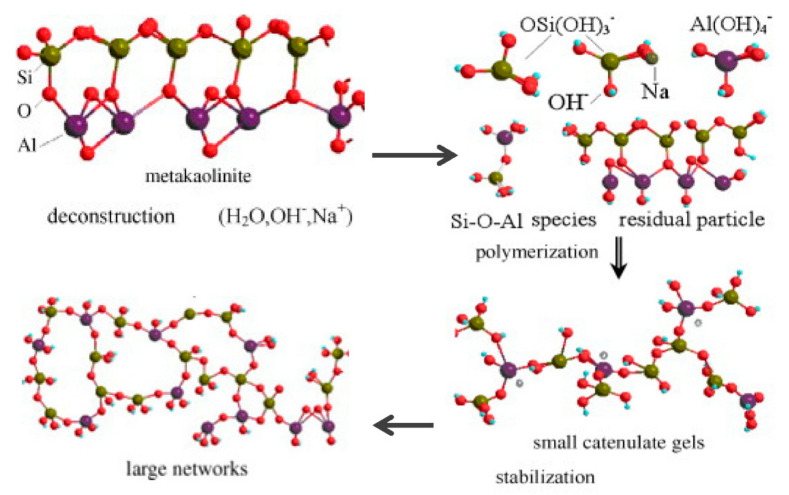
Reaction mechanism of geopolymerisation with metal kaolinite process [8,11].

**Figure 2 materials-14-04253-f002:**
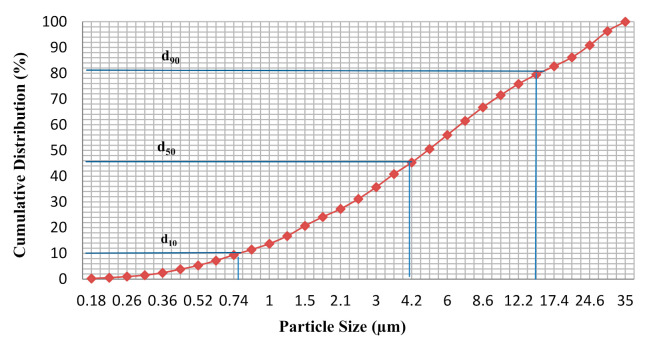
Particle size distribution of POFA.

**Figure 3 materials-14-04253-f003:**
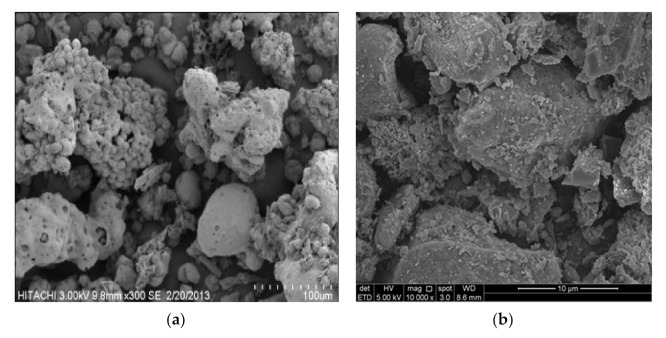
Microstructure of POFA (**a**) before [11,31] and (**b**) after grinding at 10 k magnification.

**Figure 4 materials-14-04253-f004:**
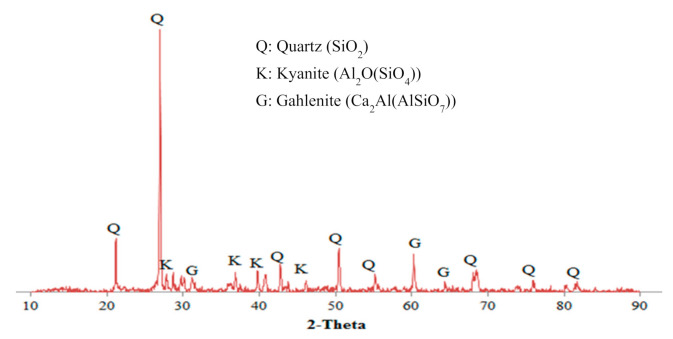
XRD pattern of POFA [32].

**Figure 5 materials-14-04253-f005:**
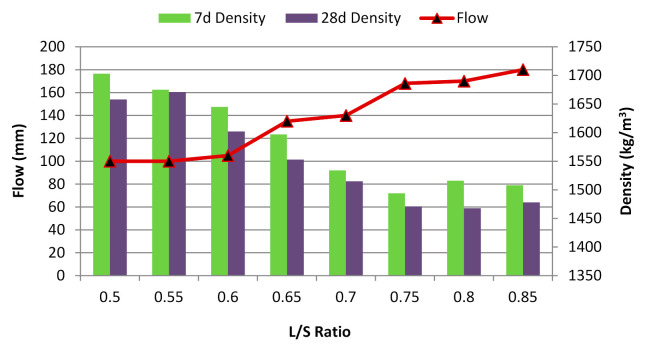
Flow of alkali-activated POFA paste at different L/S ratios.

**Figure 6 materials-14-04253-f006:**
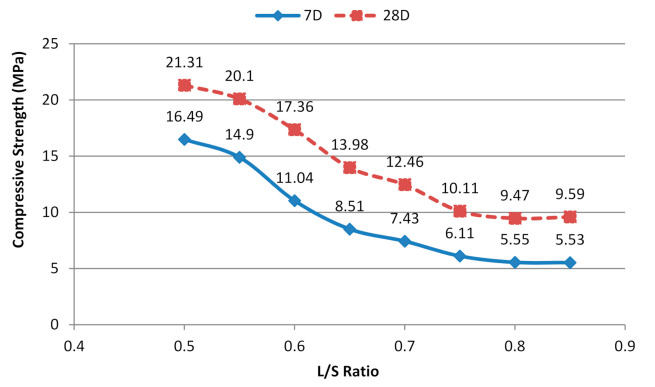
Compressive strength of alkali-activated POFA paste at different L/S ratios.

**Figure 7 materials-14-04253-f007:**
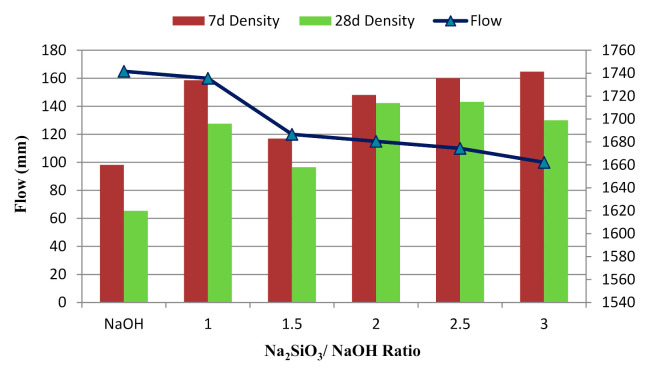
Flow of alkali-activated POFA paste at different Na_2_SiO_3_/NaOH ratios.

**Figure 8 materials-14-04253-f008:**
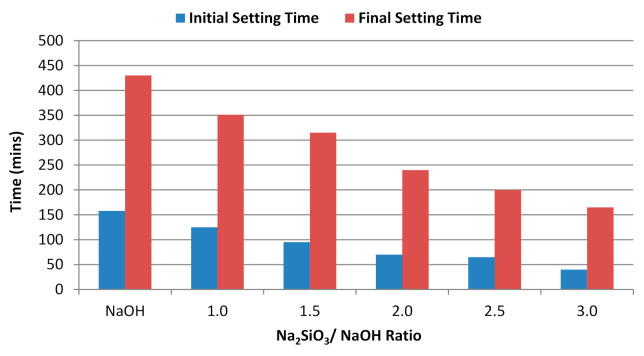
Setting time of alkali-activated POFA paste at different Na_2_SiO_3_/NaOH ratios.

**Figure 9 materials-14-04253-f009:**
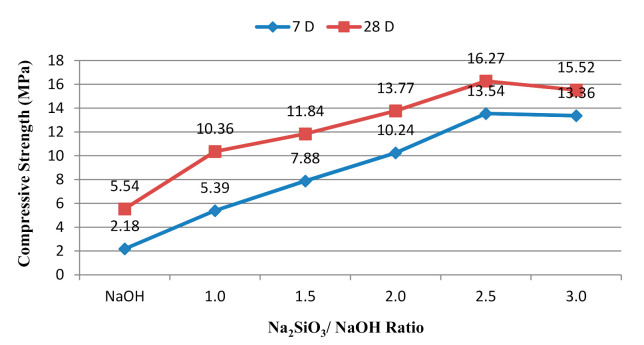
Compressive strength of alkali-activated POFA paste at different Na_2_SiO_3_/NaOH ratios.

**Figure 10 materials-14-04253-f010:**
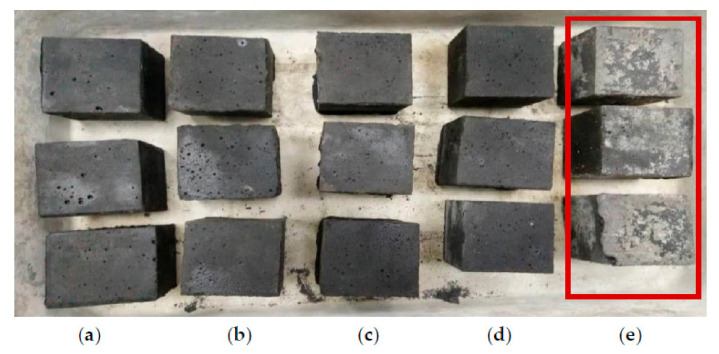
Appearance of alkali-activated POFA geopolymer at alkaline activator ratios of (**a**) 3.0, (**b**) 2.5, (**c**) 2.0, (**d**) 1.5, and (**e**) 1.0.

**Figure 11 materials-14-04253-f011:**
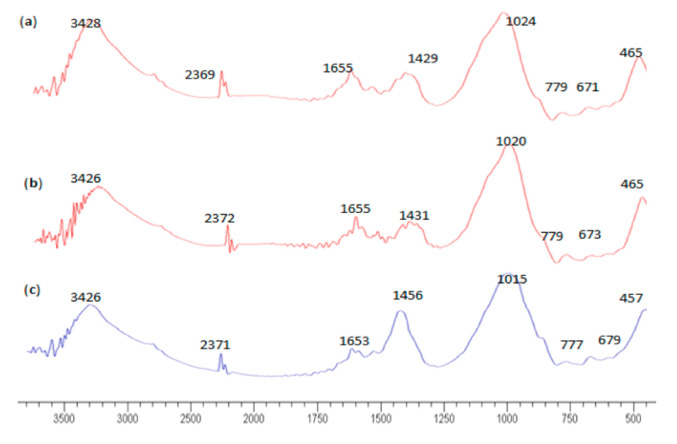
FTIR spectra for alkaline-activated POFA paste: (**a**) Na_2_SiO_3_/NaOH = 1.0, (**b**) Na_2_SiO_3_/NaOH = 2.5, and (**c**) NaOH.

**Figure 12 materials-14-04253-f012:**
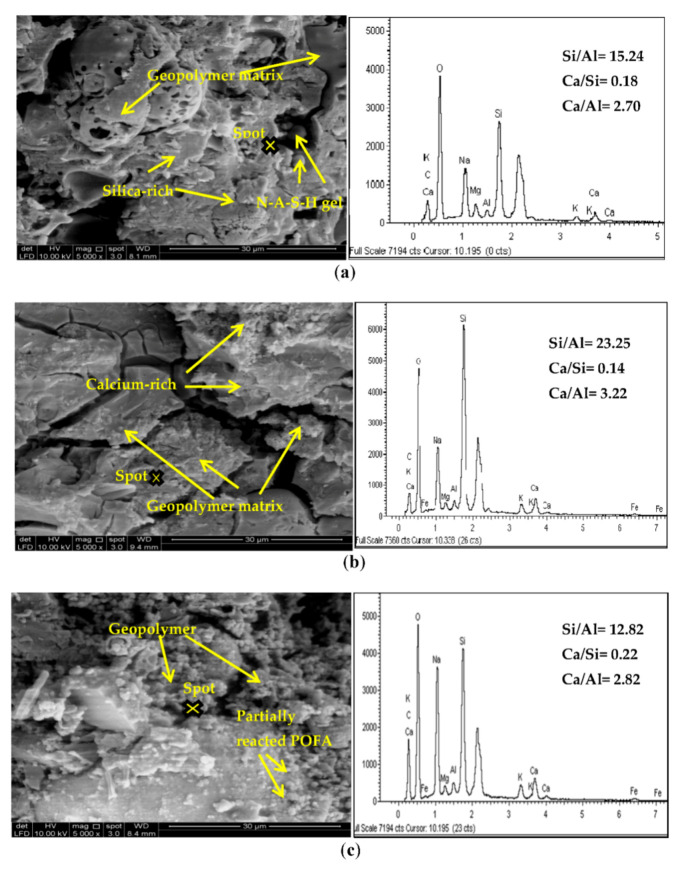
SEM images at 5000× magnification and EDX of alkaline activated POFA geopolymer: (**a**) Na_2_SiO_3_/NaOH = 1.0, (**b**) Na_2_SiO_3_/NaOH = 2.5, and (**c**) NaOH.

**Table 1 materials-14-04253-t001:** Proportions of various L/S in the mixture.

Mix	POFA (g)	Alkaline (g)	L/S
1	900	450	0.50
2	900	495	0.55
3	900	540	0.60
4	900	585	0.65
5	900	630	0.70
6	900	675	0.75
7	900	720	0.80
8	900	765	0.85

**Table 2 materials-14-04253-t002:** Mixing proportions using various alkaline activator ratios.

Mix	POFA(g)	Alkaline (g)	L/S	Na_2_SiO_3_/NaOH
1	900	540	0.6	1.0
2	900	585	0.6	1.5
3	900	630	0.6	2.0
4	900	675	0.6	2.5
5	900	720	0.6	3.0

**Table 3 materials-14-04253-t003:** Characteristic absorption peak of the FTIR spectrum.

FTIR Peak, cm^−1^			Functional Group Assigned
Na_2_SiO_3_/NaOH = 1.0	Na_2_SiO_3_/NaOH = 2.5	NaOH	
3428	3426	3426	Stretching vibration O-H
1655	1655	1653	Bending vibration H-O-H
1429	1431	1456	Asymmetric stretching vibration O-C-O
1024	1020	1015	Asymmetric stretching vibration Si-O-T
779	779	777	Bending symmetric stretching vibration Si-O-Si
671	673	679	Bending symmetric Si-O of SiO_4_
465	465	457	Bending vibration Si-O-Si and O-Si-O

## Data Availability

The data presented in this study are available on request from the corresponding author/publisher.

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
