# Peer review of "Influence of Liquid-to-Solid and Alkaline Activator (Sodium Silicate to Sodium Hydroxide) Ratios on Fresh and Hardened Properties of Alkali-Activated Palm Oil Fuel Ash Geopolymer"

_materials, 2021, doi:10.3390/ma14154253_

Round 1

Reviewer 1 Report

The paper reports the effect Liquid-to-Solid and Alkaline Activator Ratios on the properties of alkali-activated Palm Oil Fuel Ash. In my opinion, the investigation was interesting, well-designed and well-executed. Few suggestions to make the paper suitable for more readers are:             

  • Since the dominant products formed are C-S-H gel and sodium-alumino-silicate hydrates, the authors should use alkali-activated Palm Oil Fuel Ash (POFA) instead Palm Oil Fuel Ash Geopolymer
  • As the authors mentioned, the alkali-activated Palm Oil Fuel Ash (POFA) has been investigated previously. Therefore, I strongly urge the authors to enrich the introduction section with further literature review.
  • There is no XRD data reported in the manuscript. It will be important to monitor the formation of hydrate products or geopolymer using XRD. SEM/EDX does not allow quantifying of products formed.
  • Kywords: it would be better to write pofo in uppercase.
  • Conclusion: “Chemical analysis of POFA showed that the primary crystalline phase is quartz”, delete this sentence.

Reviewer 2 Report

There is still the need to improve the English of the manuscript.

Some examples:

66 "from difference materials..."

68 "of 3 needed..."

109 "The findings may promote the further investigation of the application..."

132/133 "Ordinary Portland cement (OPC) has specific surface area of 4400 cm2/g and specific gravity of 3.15." This sentence is needed? Make a proper reference. Why in cm2/g when in line 117 the value for POFA is in m2/g. Normalize to m2/kg.

139/140 "Compared with the oxide compounds existing on cement, CaO occupies the highest percentages followed by SiO2." Where in OPC or in POFA? In POFA of course, but it is a little confused.

160/161 "A total of 480 g solute was dissolved in 1 L preparation solution to produce the required concentration of NaOH."

340 "For the L/S ratio, no efflorescence was observed." what L/S ratio?

519/520 "...had the properties which were enable it to be used as a sustainable building material."

So please revise the manuscript carefully. I advise an English native speaker.

Reviewer 3 Report

Review Manuscript:

“Influence of Liquid-to-Solid and Alkaline Activator Ratios on Fresh and Hardened Properties of Palm Oil Fuel Ash Geopolymer”

The manuscript contains information on the impact that various liquid to solid ratios as well as alkaline activator ratios have on the fresh and hardened properties of POFA gopolymer. Although POFA geopolymer is limited to countries that produce palm oil the topic is quite interesting that is worth investigation. In general, the manuscript is well written and scientifically sound.

Following are some comments that should be addressed by the authors:

The yellow text in Figure 11 seems misplaced.

The title is somewhat misleading as there are several types of  alkaline activators that could potentially be used and thus the liquid to solid ratio will depend upon the type of activator. I would suggest the authors mention in the title that they used sodium hydroxide and sodium silicate.

Line 373 . You mention “For the L/S ratio, no efflorescence was observed.”, Which L/S ratio?

Line 374. You mention “The presence of efflorescence affects the strength of the geopolymer.” How do you come to this conclusion. The presence of efflorescence means that there is excess NaOH. The low compressive strength may be a result of the excessive NaOH but I wouldn’t make the statement that efflorescence affects the strength.

Figure 9. Appearance of POFA geopolymer at different alkaline activator ratios (3.0, 2.5, 2.0, 1.5 and 1.0)   -   Which sample corresponds to 3.0 and which to 1.0? Please assign letters/numbers to the samples shown in the figure.

Line 523. Replace “This study showed that the activator POFA had the properties which were enable it to be used as a sustainable building material.”

With “This study showed that the activator POFA has properties which enable it to potentially be used as a sustainable building material.”

Line 525 “The cementitious characteristic as binder to react with suitable raw would provide the ideal outcome for construction industry” please rephrase , (what do you mean by suitable raw?)

In general, the figures are well designed and quite informative.

The presentation of the results is very clear, and the discussion is interesting.

Author Response

This manuscript is a resubmission of an earlier submission. The following is a list of the peer review reports and author responses from that submission.

Round 1

Reviewer 1 Report

The authors have used Palm oil fuel ash (POFA) as raw material in the alkaline activation method, using sodium hydroxide and sodium silicate as activators. They have performed characterizations of raw material and produced samples by XRD, SEM/EDS, FTIR, workability and compressive strength at 7 and 28 days.

This is a typical study of alkaline activation using ashes as raw materials. Usually F-type fly ashes from fuel power stations or C-type ashes from wood burners are investigated. However the amount of POFA produced nowadays deserves the investigation of their use as raw material for alkaline activation.

The POFA was characterized in terms of microstructure (before and after grinding), chemical composition (by XRF) and crystalline phases (by XRD). The POFA present a medium content of CaO and a low content of Al2O3 so it must be classified (ASTM).

There is a lot of discussion if the term “Geopolymer” could be used in such context. According to Angel Palomo in the case of fly ashes the term “alkaline activation” should be used instead. Geopolymer is the term when metakaolin is used. See for example the works of Joseph Davidovits, Geopolymer Chemistry and Applications (2008) ISBN: 9782951482012 (your ref. 29) or Angel Palomo and Ana Fernandez-Jimenez, for example World of Coal Ash (WOCA) Conference - May 9-12, 2011, in Denver, CO, USA http://www.flyash.info/2011/205-Palomo-2011.pdf

The whole text must be revised carefully by an English native speaker because there are a lot of sentences that must be corrected and improved (in particular the sentences marked in yellow in the pdf file).

For final version the authors should provide figures with higher quality (at least 600 dpi for online version, although 1200 dpi is best).

1- The abstract is a little extend so it could be more concise.

2- Revise the following paragraphs of the abstract:

“However, this was not the ideal ratio since it was fast set before casting” – What is the ideal ratio and why?

“As a conclusion, alkaline activator ratio of 2.5 was the optimum value for the production…” - Explain before what is the "activator ratio" to avoid confusion with "liquid to solid ratio".

3- Remove sentence in line 97 page 3.

4- The authors must support by references the sentence on line 115-117 page 3; “It is the focus to figure out the possibility of the POFA reacts with activator additive to perform the cementitious properties.” The proper compositions for alkaline activation were also pointed out by Davidovits and by A. Palomo (and many other authors) and are based on SiO2/Al2O3, SiO2/CaO or SiO2/Na2O ratios. In addition the amorphous content (vitreous) is very important as A. Palomo claims many times.

5- The authors must include in section 2. Experimental Method the description of characterization techniques and equipment used (including the model). Something like: “the microstructures of POFA were obtained using a SEM from Hitachi model ###; the samples were observed in low vacuum mode without any conductive layer (?). The microstructures of the geopolymers were observed in a FEI model ### using LFD detector in low vacuum mode. EDS spectra were obtained using …, wit ## s of live acquisition time”.

6- In addition of the compositions of Table 1, the authors must provide the “loss on ignition” (at 900C) of the POFA because it is very important to know the content of unburnt stuff. Loss on ignition above 3% is undesired and could cause a reduction of compressive strength.

7- The authors must compare compositions from table 1 to ASTM C618-12A (or other similar).

8- Figure 3 – It’s very hard to identify Kyanite or Gehlenite from the figure provided. The authors only present one peak of each. At least 3 peaks from each phase should be indicated. However I believe that the small peak at 47° is also Kyanite and not Quartz as the authors assigned with a Q. The low quality of the figure does not allow going deep into the interpretation. In addition authors should provide references from databases (ICDD or AMCSD for example). Kyanite can be AMSCD file number 1892 and Gehlenite AMCSD file number 7694.

9- In line 127-131 the composition of the Na2SiO3 is repeated in two paragraphs. However the provided information is rather different. There is a mistake somewhere.

10- The text about setting times (line 228 page 7 up to line 243 page 8) is not in accordance with Figure 7. Revise all the text.

11- Line 247-248 page 8: “It was due to the content of sodium silicate which acted as the coagulate role to accelerate the dissolution during polymerization”. In my opinion the main role of sodium silicate is to provide soluble silicon to the alkaline activation (see Figure 1), because the rate of silicon dissolution is much slower when compared to the rate of aluminum dissolution. See the following works:

- García-Lodeiro I, Fernandez-Jimenez A, Palomo A. Variation in hybrid cements

over time. Alkaline activation of fly asheportland cement blends. Cem Concr

Res 2013;52:112e22.

- Cheng H, Lin K-L, Cui R, Hwang C-L, Chang Y-M, Cheng T-W. The effects of

SiO2/Na2O molar ratio on the characteristics of alkali-activated waste catalystemetakaolin based geopolymers. Constr Build Mater 2015;95:710e20.

12- Figure 8 should not be in “line+markers” type but in “column” type, like figure 7. In alternative it could be in “line+markers” type if “NaOH” is represented by “0” i.e. the XX axis is numeric.

13- Text from line 291-295 and line 302-303 (page 9) must be revised. FTIR Kyanite spectra can be found in https://rruff.info/Kyanite/R040119. Main peak is around 920 cm-1.

14- Figure 10. Peak around 2370 is from atmospheric CO2. Remove from peak label in the graph.

15- Line 330 page 11. “…with lower spread of particle size distribution…” The authors must complete the characterization of POFA including a graph with particle size distribution before and after grinding. This graph must be included in the beginning of section 2.1 Materials together with Figure 2.

16- Line 332 page 11. “…higher specific surface…”. The authors must include a measurement of specific surface before and after grinding.

17- Line 339-340. “There was a discernible formation of the crystalline compound within the binding gel in the NaOH activated binder system.”. Crystalline compound? What crystalline compound? Where in the figure?

18- Line 351. “Morphologically, this phase was similar to…” What phase?

19- Line 668-376 page 12. This is obvious and can be shortened. However different compositions in different samples can be commented in terms of added NaOH and Na2SiO3.

20- Line 377-399 pages 12-13. The SiO2/Al2O3, SiO2/Na2O, SiO2/CaO ratios are very important aspects of geopolymerization. Revise the text because it is very confusing. Make calculations of the theoretical ratios of different samples taking into account the composition and amount of POFA, NaOH and Na2SiO3 added to make each sample. Correlated the results with compressive strength.

21- Revise figure 13 to oxides (not elements) and discuss properly in the text.

22- Conclusion 4 is not supported by any data presented in the text.

23- Revise the text in conclusion 5.

24- Revise conclusion 6 according to the results of the new text written as a consequence of comments 20 and 21.

25- Include missing pages in the references. Check carefully that references are in accordance with “Materials” standards.
